# Influence of Source Parameters and Non-Kolmogorov Turbulence on Evolution Properties of Radial Phased-Locked Partially Coherent Vortex Beam Array

**Jiao Wang** [1,2,]*[ ], **Mingjun Wang** [3], **Sichen Lei** [3], **Zhenkun Tan** [4], **Chenbai Wang** [1,2] and **Yuanfei Wang** [5]

1 School of Electronic Information and Artificial Intelligence, Shaanxi University of Science and Technology, Xi'an 710021, China; wcb000402@163.com
2 Shaanxi Joint Laboratory of Artificial Intelligence, Shaanxi University of Science and Technology, Xi'an 710021, China
3 Faculty of Automation & Information Engineering, Xi'an University of Technology, Xi'an 710048, China; wangmingjun@xaut.edu.cn (M.W.); lsc429@163.com (S.L.)
4 Faculty of Optoelectronic Engineering, Xi'an Technological University, Xi'an 710021, China; tzk0828@163.com
5 Sinochem Lantian Fluoro Materials Co., Ltd., Shaoxing 312369, China; wangyuanfei@sinochem.com
* Correspondence: wangj_922@163.com; Tel.: +86-150-2927-0695

**Abstract:** Partially coherent optical vortices have been applicated widely to reduce the influence of atmospheric turbulence, especially for free-space optical (FSO) communication. Furthermore, the beam array is an effective way to increase the power of the light source, and can increase the propagation distance of the FSO communication system. Herein, we innovatively report evolution properties of the radial phased-locked partially coherent vortex (RPLPCV) beam array in non-Kolmogorov turbulence. The analytical expressions for the cross-spectral density and the average intensity of an RPLPCV beam array propagated through non-Kolmogorov turbulence are obtained. The numerical results reveal that the intensity distribution of the RPLPCV array propagated in the non-Kolmogorov turbulence is gradually converted to a standard Gaussian distribution. In addition, the larger the radial radius, radial number and waist radius are, the smaller the coherence length is. Moreover, the longer the wavelength is, the shorter the propagation distance required for the intensity distribution of the RPLPCV beam array to be converted into a Gaussian distribution in the non-Kolmogorov turbulence. The research in this paper provides a theoretical reference for the selection of light sources and the suppression of turbulence effects in wireless optical communication.

**Keywords:** intensity distribution; partially coherent optical vortices; beam array; non-Kolmogorov turbulence

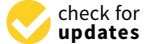



## 1. Introduction

During the past several decades, emerged insight into practical applications of free-space optical (FSO) communications [1–3] and quantum information [4,5] has culminated in the concept of the vortex beams [6]. Among the many models of vortex beams, the partially coherent vortex beam combines the respective properties of vortex beams and partially coherent beams [7] to show unique advantages, such as smaller beam expansion, a stronger self-healing ability, etc., attracting widespread attention [8–10]. In FSO communication systems, in addition to beam combining technology (BCT) [11] to increase the output of high-power lasers, using partially coherent vortex beams as carriers is also an effective way to increase the transmission distance and reduce atmospheric turbulence effects. Therefore, combining the BCT and the inherent properties of partially coherent vortex beams, studying the propagation characteristics of the radial phased-locked partially coherent vortex beam array in non-Kolmogorov turbulence has a certain theoretical basis and technical support for the field of information transmission.

In 2006, Duan and Lü et al. [12,13] introduced a new form of laser beam called Four-petal Gaussian (FPG) beam, and discussed the propagation properties of FPG beams in free space and atmospheric turbulence [14–16]. The study found that the intensity distribution form of the FPG beam did not change within a short propagation distance, but with the increase in the propagation distance, the intensity distribution of the FPG beam gradually expands to a Gaussian distribution [17,18]. The related research of the FPG beam laid the foundation for the proposal of various types of beam arrays and the research of their propagation properties. Naturally, all kinds of beam arrays have been received much attention, such as stochastic Gaussian–Schell model array beams [19], rectangular Lorentz beam array [20], radial phased-locked Lorentz beam array [21], radial Gaussian beam array [22], radial Gaussian Schell-model array beams [23], radial phased-locked partially coherent anomalous hollow [24], flat-topped beam array [25], partially coherent Airy beam arrays [26] and so on.

In 2015, Singh et al. [27] proposed and experimentally demonstrated a vortex array embedded in a partially coherent beam, and confirmed the existence of the vortex array by presenting amplitude and phase distributions of the complex coherence function. Accordingly, Liu et al. [28] discussed the properties of spreading of a radial phased-locked partially coherent flat-topped vortex beam array propagating through non-Kolmogorov medium. However, to the best of our knowledge, less work has been reported regarding the evolution properties of the radial phased-locked partially coherent vortex beam array in non-Kolmogorov turbulence. In this paper, based on the partially coherent vortex beam, a mathematical model of the radial phased-locked partially coherent vortex (RPLPCV) beam array is established. Using numerical analysis, the evolution law of the RPLPCV beam array propagated in non-Kolmogorov turbulence is discussed, and the influence of source parameters and atmospheric turbulence parameters on its intensity distribution is analyzed in detail. It is believed that the results will benefit the study of fourth-order moment properties of RPLPCV beam arrays.

## 2. Theoretical Formulation

When the spiral phase factor $\exp(-il\theta)$ of a vortex beam is equivalent to $[r_x - i\,\mathrm{sgn}(l)\,r_y]^{|l|}$ in a Cartesian coordinate system [29,30], where $l$ is the azimuthal index (also called the topological charge) and $\theta$ is the azimuthal coordinates in cylindrical coordinates, the cross-spectral density function (CSDF) of a partially coherent vortex beam [31] at a pair of points with arbitrary transverse position vectors $\mathbf{r}_1 = (r_{1x}, r_{1y})$ and $\mathbf{r}_2 = (r_{2x}, r_{2y})$ in the source plane $z = 0$ takes the form:

$$W_0^{(1)}(\mathbf{r}_1, \mathbf{r}_2; 0) = \exp\left(-\frac{\mathbf{r}_1^2 + \mathbf{r}_2^2}{4w_0^2}\right) \exp\left(-\frac{|\mathbf{r}_1 - \mathbf{r}_2|^2}{2\delta^2}\right) \left[r_{1x} - i\mathrm{sgn}(l)r_{1y}\right]^{|l|} \left[r_{2x} + i\mathrm{sgn}(l)r_{2y}\right]^{|l|} \quad (1)$$

where $w_0$ is the beam radius of the Gaussian beam at the waist, $\delta$ is the coherence length of the source and $\mathrm{sgn}(\cdot)$ is a symbolic function.

According to Equation (1) and the characteristics of the radial phased-locked beam array (as shown in Figure 1), the RPLPCV beam array of $N$ beamlets in the source plane $z = 0$ can be expressed as:

$$W_0^{(N)}(\mathbf{r}_1, \mathbf{r}_2; 0) = \sum_{n=1}^{N} \exp\left[-\frac{(\mathbf{r}_1 - \mathbf{r}_n)^2 + (\mathbf{r}_2 - \mathbf{r}_n)^2}{4w_0^2}\right] \exp\left(-\frac{|\mathbf{r}_1 - \mathbf{r}_2|^2}{2\delta^2}\right)$$
$$\times \left[(r_{1x} - r_{nx}) - i\mathrm{sgn}(l)(r_{1y} - r_{ny})\right]^{|l|} \left[(r_{2x} - r_{nx}) + i\mathrm{sgn}(l)(r_{2y} - r_{ny})\right]^{|l|} \quad (2)$$

where $N$ is the beamlet number and $\mathbf{r}_n = (r_{nx}, r_{ny})$ is the transverse position vector of the center of the $n$-th beamlet located at the source plane (i.e., the radial radius vector), $n = 1, 2, 3, \ldots, N$. Because the spiral phase factor in Equation (2) cannot be integrated and an analytic expression cannot be given when the spiral phase factor $[(r_x - r_{nx}) - i\,\mathrm{sgn}(l)(r_y - r_{ny})]^{|l|}$ is expanded, in the following, we mainly study the evolution properties of an RPLPCV beam array with the topological charge $l = 1$ in non-Kolmogorov turbulence. Then, Equation (2) can be written as:

$$W_0^{(N)}(\mathbf{r}_1, \mathbf{r}_2; 0) = \sum_{n=1}^{N} \exp\left[-\frac{(\mathbf{r}_1 - \mathbf{r}_n)^2 + (\mathbf{r}_2 - \mathbf{r}_n)^2}{4w_0^2}\right] \exp\left(-\frac{|\mathbf{r}_1 - \mathbf{r}_2|^2}{2\delta^2}\right)$$

$$\times \left\{ r_{1x}r_{2x} + r_{1y}r_{2y} - (r_{1x} + r_{2x}) - (r_{1y} + r_{2y})r_{ny} + r_{nx}^2 + r_{ny}^2 + i\left[r_{1x}r_{2y} - r_{2x}r_{1y} + (r_{1y} - r_{2y})r_{nx} - (r_{1x} - r_{2x})r_{ny}\right] \right\} \quad (3)$$

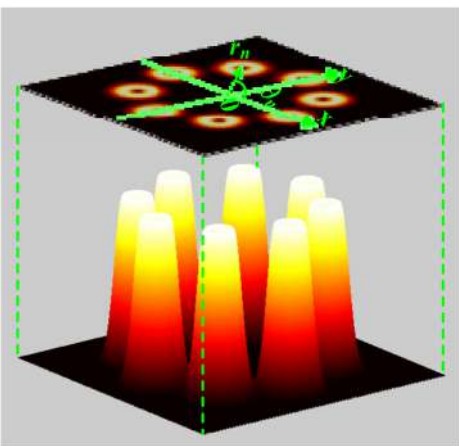

**Figure 1.** Schematic diagram of an RPLPCV beam array with $N$ = 8 as an example.

According to the extended Huygens–Fresnel principle [32], the CSDF of the RPLPCV beam array propagated close to the positive $z$-axis into the half-space $z > 0$ is given by:

$$W^{(N)}(\boldsymbol{\rho}_1, \boldsymbol{\rho}_2; z) = \left(\frac{k}{2\pi z}\right)^2 \int \int \sum_{n=1}^{N} \exp\left[-\frac{(\mathbf{r}_1 - \mathbf{r}_n)^2 + (\mathbf{r}_2 - \mathbf{r}_n)^2}{4w_0^2}\right] \exp\left(-\frac{|\mathbf{r}_1 - \mathbf{r}_2|^2}{2\delta^2}\right)$$

$$\times \left\{ r_{1x}r_{2x} + r_{1y}r_{2y} - (r_{1x} + r_{2x})r_{nx} - (r_{1y} + r_{2y})r_{ny} + r_{nx}^2 + r_{ny}^2 + i\left[r_{1x}r_{2y} - r_{2x}r_{1y} + (r_{1y} - r_{2y})r_{nx} - (r_{1x} - r_{2x})r_{ny}\right] \right\} \quad (4)$$

$$\times \exp\left[-\frac{ik}{2z}\left(|\boldsymbol{\rho}_1 - \mathbf{r}_1|^2 - |\boldsymbol{\rho}_2 - \mathbf{r}_2|^2\right)\right] \langle\exp[\psi(\boldsymbol{\rho}_1, \mathbf{r}_1) + \psi^*(\boldsymbol{\rho}_2, \mathbf{r}_2)]\rangle d\mathbf{r}_1 d\mathbf{r}_2$$

where $\boldsymbol{\rho}_1 = (\rho_1, \varphi_1)$ and $\boldsymbol{\rho}_2 = (\rho_2, \varphi_2)$ are a pair of points with arbitrary transverse position vectors in the receiver plane, and $k$ is the optical wave number related to the wavelength $\lambda$ by $k = 2\pi/\lambda$. $\psi(\boldsymbol{\rho}, \mathbf{r})$ stands for the random part of the complex phase of a spherical wave due to the atmospheric turbulence from the point $(\mathbf{r}, 0)$ to $(\boldsymbol{\rho}, z)$. The angular brackets denote averaging over the ensemble of turbulence media, which can be expressed as [33]:

$$\langle\exp[\psi(\boldsymbol{\rho}_1, \mathbf{r}_1) + \psi^*(\boldsymbol{\rho}_2, \mathbf{r}_2)]\rangle = \exp\left[-\frac{1}{2}D_\psi(\mathbf{r}_1 - \mathbf{r}_2)\right] = \exp\left[-\frac{M}{2}(\mathbf{r}_1 - \mathbf{r}_2)^2\right] \quad (5)$$

where $D_\psi(\mathbf{r}_1 - \mathbf{r}_2)$ is the wave structure function, and when we analyze the influence of the inner and outer scales on the polarization properties using the modified von Karman spectrum model, the parameter $M$ is:

$$M = 2/3\pi^2 k^2 z A(\alpha) \int_0^\infty \kappa^3 C_0 \exp(-\kappa^2/\kappa_m^2)(\kappa^2 + \kappa_0^2)^{-\alpha/2} d\kappa \quad (6)$$

where $\alpha$ is generalized exponent factor in the range of $3 < \alpha < 4$, $A(\alpha)$ is generalized amplitude, $\kappa_m = c(\alpha)/l_0$, $\kappa_0 = 2\pi/L_0$, and $l_0$ and $L_0$ represent the inner scale and outer scale, respectively. $C_0$ is the near-surface refractive index structure constant of turbulence atmosphere. The expressions of $A(\alpha)$ and $c(\alpha)$ are:

$$\begin{cases} A(\alpha) = \frac{1}{4\pi^2}\Gamma(\alpha - 1)\cos\left(\frac{\pi}{2}\alpha\right) \\ c(\alpha) = \left[\frac{2\pi}{3}\Gamma\left(5 - \frac{\alpha}{2}\right)A(\alpha)\right]^{1/(\alpha-5)} \end{cases} \quad (7)$$

Substituting Equations (5) and (6) into Equation (4), we obtain the following expression:

$$
\begin{aligned}
W^{(N)}(\boldsymbol{\rho}_1,\boldsymbol{\rho}_2;z) =& \left(\frac{k}{2\pi z}\right)^2 \exp\left[-\frac{ik}{2z}\left(\rho_1^2-\rho_2^2\right)\right]\sum_{n=1}^{N}\exp\left(-\frac{1}{2w_0^2}\mathbf{r}_n^2\right)\int \exp\left(-E\mathbf{r}_d^2\right)\exp\left[\frac{ik}{2z}(\boldsymbol{\rho}_1+\boldsymbol{\rho}_2)\cdot\mathbf{r}_d\right]\exp\left(-\frac{ik}{z}\mathbf{r}_d\cdot\mathbf{r}_c\right)d\mathbf{r}_d \\
&\times\int \exp\left(-\frac{1}{2w_0^2}\mathbf{r}_c^2\right)\exp\left(\frac{1}{w_0^2}\mathbf{r}_n\cdot\mathbf{r}_c\right)\exp\left[\frac{ik}{z}(\boldsymbol{\rho}_1-\boldsymbol{\rho}_2)\cdot\mathbf{r}_c\right]d\mathbf{r}_c \\
&\times\left[\left(\mathbf{r}_c^2-2r_{cx}r_{nx}-2r_{cy}r_{ny}+\mathbf{r}_n^2\right)+\left(-\tfrac{1}{4}\mathbf{r}_d^2+ir_{dy}r_{ny}-ir_{dx}r_{ny}\right)+ir_{cy}r_{dx}-ir_{cx}r_{dy}\right]
\end{aligned}
\tag{8}
$$

Equation (8) is a combination of nine terms, so each integral must be solved separately. Where:

$$
E=\frac{1}{8w_0^2}+\frac{1}{2\delta^2}+\frac{M}{2}
\tag{9}
$$

$$
\begin{cases} \mathbf{r}_c=\frac{1}{2}(\mathbf{r}_1+\mathbf{r}_2) \\ \mathbf{r}_d=\mathbf{r}_1-\mathbf{r}_2 \end{cases},\quad \begin{cases} \boldsymbol{\rho}_c=\frac{1}{2}(\boldsymbol{\rho}_1+\boldsymbol{\rho}_2) \\ \boldsymbol{\rho}_d=\boldsymbol{\rho}_1-\boldsymbol{\rho}_2 \end{cases}
\tag{10}
$$

The variables $\mathbf{r}_c$ and $\mathbf{r}_d$ are sequentially integrated with the help of the following integral formulae:

$$
\int_{-\infty}^{+\infty}\left[e^{-ax^2}\right]e^{-i2\pi w_x x}dx\cdot\int_{-\infty}^{+\infty}\left[e^{-ay^2}\right]e^{-i2\pi w_y y}dy=\frac{\pi}{a}\cdot e^{-\frac{\pi^2 w^2}{a}}
\tag{11}
$$

$$
\int_{-\infty}^{\infty}\exp\left(-px^2\pm qx\right)dx=\sqrt{\frac{\pi}{p}}\exp\left(\frac{q^2}{4p}\right)[p>0]
\tag{12}
$$

$$
\int_{-\infty}^{\infty}x^n\exp\left(-px^2+2qx\right)dx=(2i)^{-n}\frac{\sqrt{\pi}}{\left(\sqrt{p}\right)^{n+1}}\exp\left(\frac{q^2}{p}\right)\mathrm{H}_n\left[i\left(\frac{q}{\sqrt{p}}\right)\right]
\tag{13}
$$

where $\mathrm{H}_n(\cdot)$ denotes the Hermite polynomial of mode order $n$. We obtain the following expression for the CSDF of the RPLPCV beam array in the receiver plane:

$$
\begin{aligned}
W^{(N)}(\boldsymbol{\rho}_1,\boldsymbol{\rho}_2,z)=&-\frac{\pi^2 AB_1}{E}\sum_{n=1}^{N}L_1\left\{\frac{H_2[Q_{1x}]+H_2[Q_{1y}]}{4D^2}-i\frac{r_{nx}H_1[Q_{1x}]+r_{ny}H_1[Q_{1y}]}{D^{3/2}}-\frac{\mathbf{r}_n^2}{D}\right\} \\
&+\pi^2 w_0^2 AB_2\sum_{n=1}^{N}L_2\left\{\frac{H_2[Q_{1x}]+H_2[Q_{1y}]}{8C^2}+\frac{r_{nx}H_1[Q_{2y}]-r_{ny}H_1[Q_{2x}]}{C^{3/2}}\right\} \\
&-\frac{i\pi^2 w_0 A}{2\sqrt{2E}CD}\sum_{n=1}^{N}\exp\left(-\frac{1}{2w_0^2}\mathbf{r}_n^2\right)\left\{B_{1y}B_{2x}P_{1x}P_{2y}H_1[Q_{2x}]H_1[Q_{1y}]-B_{1x}B_{2y}P_{1y}P_{2x}H_1[Q_{2y}]H_1[Q_{1x}]\right\}
\end{aligned}
\tag{14}
$$

where:

$$
A=\left(\frac{k}{2\pi z}\right)^2\exp\left[-\frac{ik}{2z}\left(\rho_1^2-\rho_2^2\right)\right]
\tag{15}
$$

$$
B_1=\exp\left[-\frac{k^2}{16Ez^2}(\boldsymbol{\rho}_1+\boldsymbol{\rho}_2)^2\right]\exp\left[\frac{1}{4D}(F_1\boldsymbol{\rho}_1+F_2\boldsymbol{\rho}_2)^2\right],\quad B_2=\exp\left[-\frac{k^2 w_0^2}{2z^2}(\boldsymbol{\rho}_1-\boldsymbol{\rho}_2)^2\right]\exp\left[\frac{1}{4C}(G_1\boldsymbol{\rho}_1+G_2\boldsymbol{\rho}_2)^2\right]
\tag{16}
$$

$$
C=E+\frac{k^2 w_0^2}{2z^2},\ G_1=\frac{ik}{2z}+\frac{k^2 w_0^2}{z^2},\ G_2=\frac{ik}{2z}-\frac{k^2 w_0^2}{z^2},\ D=\frac{1}{2w_0^2}+\frac{k^2}{4Ez^2},\ F_1=\frac{k^2}{4Ez^2}+\frac{ik}{z},\ F_2=\frac{k^2}{4Ez^2}-\frac{ik}{z}
\tag{17}
$$

$$
\begin{aligned}
L_1&=\exp\left[-\left(\frac{1}{2w_0^2}-\frac{1}{4Dw_0^4}\right)\mathbf{r}_n^2\right]\exp\left[\frac{1}{2Dw_0^2}\mathbf{r}_n\cdot(F_1\boldsymbol{\rho}_1+F_2\boldsymbol{\rho}_2)\right], \\
L_2&=\exp\left[-\left(\frac{k^2}{4Cz^2}\right)\mathbf{r}_n^2\right]\exp\left[\frac{ik}{z}\mathbf{r}_n\cdot(\boldsymbol{\rho}_1-\boldsymbol{\rho}_2)\right]\exp\left[-\frac{ik}{2Cz}\mathbf{r}_n\cdot(G_1\boldsymbol{\rho}_1+G_2\boldsymbol{\rho}_2)\right]
\end{aligned}
\tag{18}
$$

$$B_{1j} = \exp\left[-\frac{k^2}{16Ez^2}\left(\rho_{1j} + \rho_{2j}\right)^2\right] \exp\left[\frac{1}{4D}\left(F_1\rho_{1j} + F_2\rho_{2j}\right)^2\right]$$

$$B_{2j} = \exp\left[-\frac{w_0^2 k^2}{2z^2}\left(\rho_{1j} - \rho_{2j}\right)^2\right] \exp\left[\frac{1}{4C}\left(G_1\rho_{1j} + G_2\rho_{2j}\right)^2\right] \quad , j = x, y \tag{19}$$

$$Q_{1j} = \frac{i}{2\sqrt{D}w_0^2}r_{nj} + \frac{i}{2\sqrt{D}}\left(F_1\rho_{1j} + F_2\rho_{2j}\right), \quad Q_{2j} = \frac{k}{2z\sqrt{C}}r_{nj} + \frac{i}{2\sqrt{C}}\left(G_1\rho_{1j} + G_2\rho_{2j}\right) \; j = x, y \tag{20}$$

$$P_{1j} = \exp\left[\left(\frac{1}{2w_0^2} - \frac{k^2}{4Cz^2}\right)r_{nj}^2\right] \exp\left[\frac{ik}{z}r_{nj}(\rho_{1j} - \rho_{2j})\right] \exp\left[-\frac{ik}{2Cz}r_{nj}(G_1\rho_{1j} + G_2\rho_{2j})\right]$$

$$P_{2j} = \exp\left(\frac{1}{4Dw_0^4}r_{nj}^2\right) \exp\left[\frac{1}{2Dw_0^2}r_{nj}(F_1\rho_{1j} + F_2\rho_{2j})\right] \quad , j = x, y \tag{21}$$

The intensity distribution of the RPLPCV beam array in the receiver plane is given by:

$$I^{(N)}(\boldsymbol{\rho}, z) = W^{(N)}(\boldsymbol{\rho}, \boldsymbol{\rho}, z) \tag{22}$$

Substituting Equations (14)–(21) into Equation (22), the intensity distribution of the RPLPCV beam array propagating through non-Kolmogorov turbulence can be obtained. Therefore, we can analyze the influence of the source parameters and non-Kolmogorov turbulence on the intensity distribution of the RPLPCV beam array.

## 3. Results

In this section, by using Equations (14)–(22), the intensity of the RPLPCV beam array can be calculated by simulation. The calculation parameters are $\lambda = 632.8$ nm, $w_0 = 10$ mm, $\delta = 2w_0$, $l = 1$, $N = 8$, $r_n = 10w_0$, $\alpha = 3.5$, $C_0 = 1.7 \times 10^{-14}$ m$^{3-\alpha}$, $z = 2$ km, $l_0 = 0.01$ m and $L_0 = 100$ m, unless other values are specified in the figure captions. We numerically study the effects of the array parameters, single source parameters and atmospheric turbulence parameters on the intensity distributions of the RPLPCV beam array propagating through non-Kolmogorov turbulence. Thereby, the evolution law of the RPLPCV beam array in non-Kolmogorov turbulence is further determined.

### 3.1. Analysis of the Influence of Beam Array Parameters

The radial radius and beamlet number are two important parameters of radial array, and the influence on the intensity distributions of the RPLPCV beam array propagated in non-Kolmogorov turbulence are shown in Figures 2–4.

Figure 2 presents the intensity distributions of the RPLPCV beam array propagated through non-Kolmogorov turbulence for different propagation distances $z = 0.5$ km, $z = 1$ km, $z = 1.5$ km, $z = 2$ km and $z = 3$ km, when the radial radii are $r_n = 5w_0$, $r_n = 7w_0$ and $r_n = 10w_0$, respectively. Combining with Figure 2a1–e1, Figure 2a2–e2 and Figure 2a3–e3, the evolution law of the intensity distribution of the RPLPCV beam array propagated along through the non-Kolmogorov turbulence is obtained. That is, with the increase in the propagation distance, the intensity distribution of the RPLPCV beam array is first converted from the array distribution to the ring distribution, and then gradually to the Gaussian-like distribution until finally to the standard Gaussian distribution. This result occurs because the central dark spot of the RPLPCV beam array's each beamlet gradually disappears with the propagation distance increase. In addition, the on-axis intensity of the RPLPCV beam array propagated through non-Kolmogorov turbulence increases as the propagation distance increases (a more detailed representation is shown in Figure 3). Comparing Figure 2a1–e1, Figure 2a2–e2 and Figure 2a3–e3, it can be found that the larger the radial radius is, the longer the propagation distance required to convert the intensity distribution of the RPLPCV beam array into the standard Gaussian distribution is. Of course, the same atmospheric turbulence environment, when the propagation distance is increased to a certain value, the larger the radial radius is, the larger the radius of the intensity distribution is, and therefore, the more serious the beam spreading is. This is

because the larger the radial radius is, the more the RPLPCV beam array is affected by non-Kolmogorov turbulence.

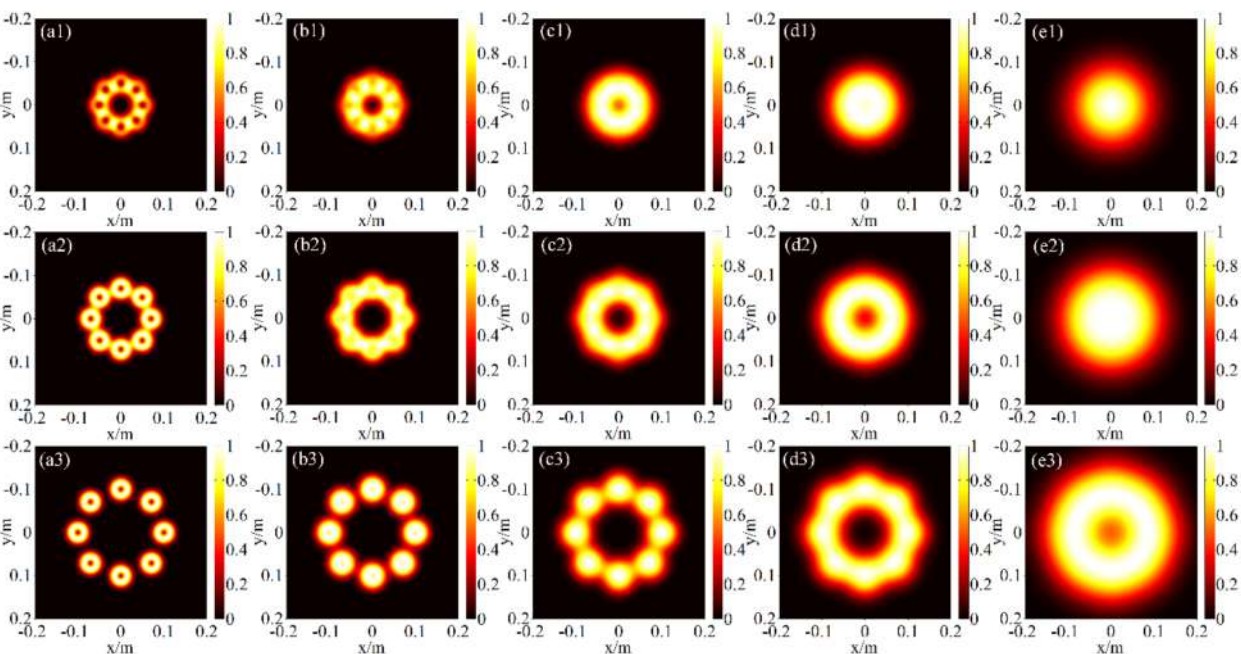

**Figure 2.** Intensity distributions of the RPLPCV beam array with different radial radii (**a1–e1**) $r_n = 5w_0$, (**a2–e2**) $r_n = 7w_0$ and (**a3–e3**) $r_n = 10w_0$ propagated through non-Kolmogorov turbulence for different propagation distances: (**a1–a3**) $z = 0.5$ km; (**b1–b3**) $z = 1$ km; (**c1–c3**) $z = 1.5$ km; (**d1–d3**) $z = 2$ km; (**e1–e3**) $z = 3$ km.

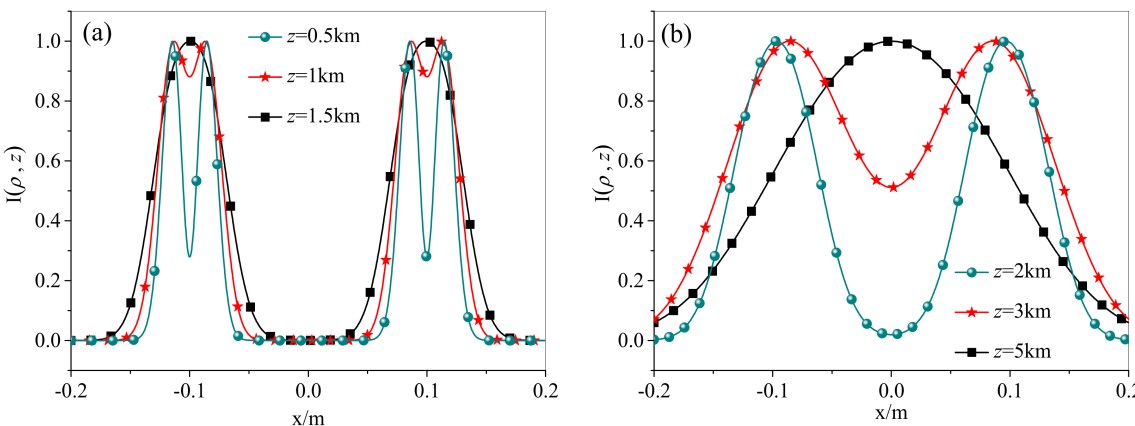

**Figure 3.** Intensity distribution profiles of the RPLPCV beam array propagated through non-Kolmogorov turbulence for different propagation distances: (**a**) $z = 0.5$ km, 1 km, 1.5 km; (**b**) $z = 2$ km, 3 km, 5 km.

Figure 4 presents the intensity distributions of the RPLPCV beam array propagated through non-Kolmogorov turbulence for different beamlet number $N = 5, 6, 7, 8, 9, 10$. It can be seen that under the same other conditions, the larger the beamlet number $N$ is, the closer the intensity distribution of the RPLPCV beam array propagated through non-Kolmogorov turbulence is to the circular distribution. This is because when the beamlet number $N$ is larger, there are more overlapping areas of adjacent beamlets. In addition, with the increase in the beamlet number $N$, the dark area at the center of the intensity distribution gradually changes from an N-sided polygon to a circle. Through the above analysis, it can be predicted that the larger the beamlet number $N$, the shorter the propagation distance required to convert the intensity distribution of the RPLPCV beam array into the standard Gaussian distribution.

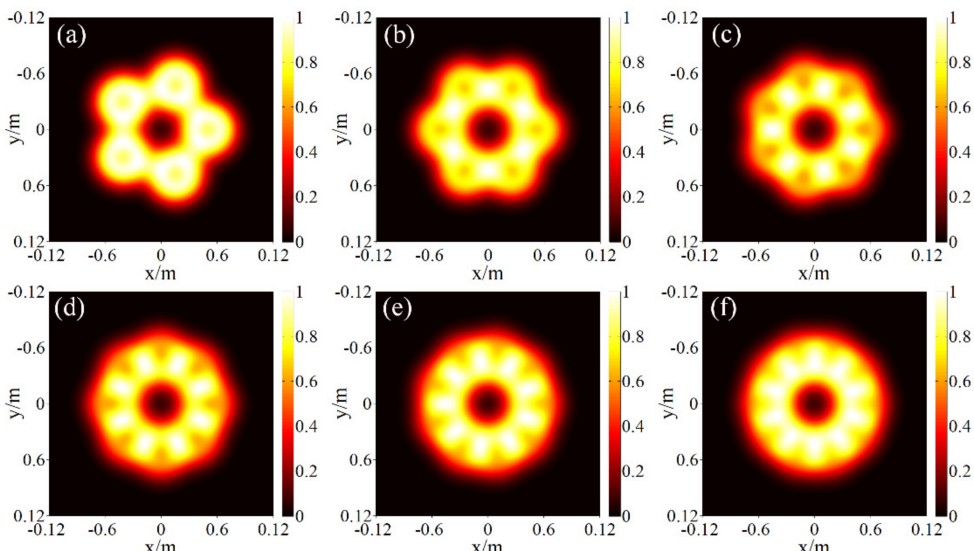

**Figure 4.** Intensity distributions of the RPLPCV beam array propagated through non-Kolmogorov turbulence for different beamlet numbers: (**a**) $N = 5$; (**b**) $N = 6$; (**c**) $N = 7$; (**d**) $N = 8$; (**e**) $N = 9$; (**f**) $N = 10$.

### 3.2. Analysis of the Influence of Single Source Parameters

In addition to the influence of the radial radius and beamlet number on the evolution properties of the RPLPCV beam array in non-Kolmogorov turbulence, the single source parameters that make up the array are also important influencing factors, such as wavelength, waist radius and coherence length. Therefore, the effects of single light source parameters on the intensity distribution of the RPLPCV beam array propagated in non-Kolmogorov turbulence are shown in Figures 5 and 6.

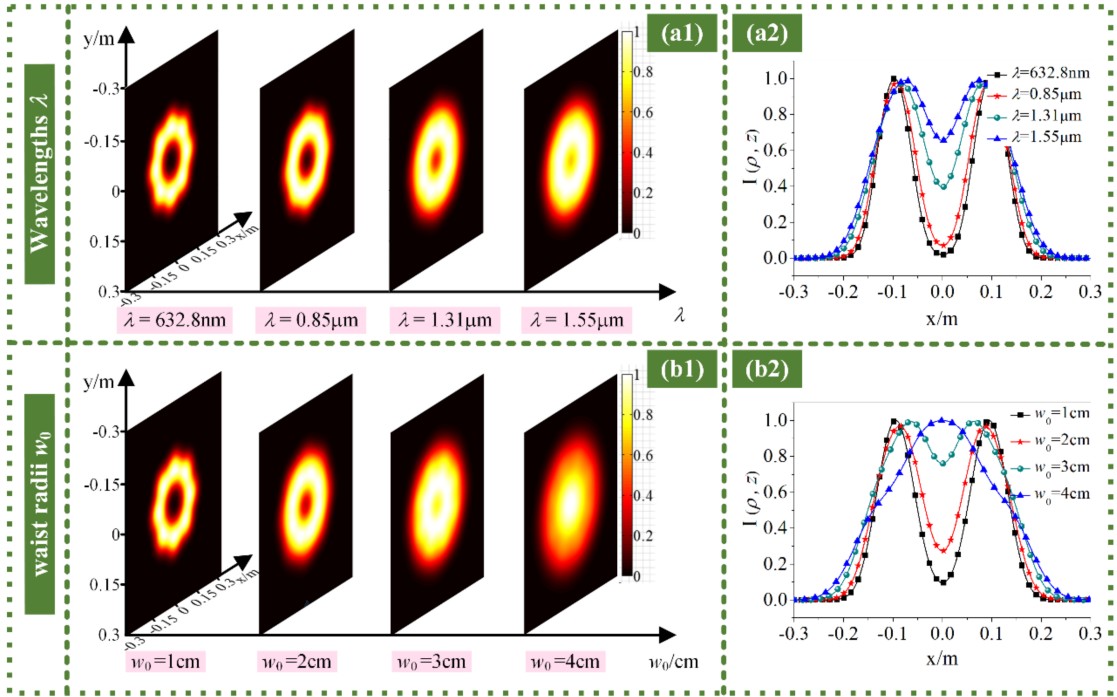

**Figure 5.** Intensity (**a1,b1**) distributions and (**a2,b2**) profiles of the RPLPCV beam array propagated through non-Kolmogorov turbulence for different (**a**) wavelengths; (**b**) waist radii.

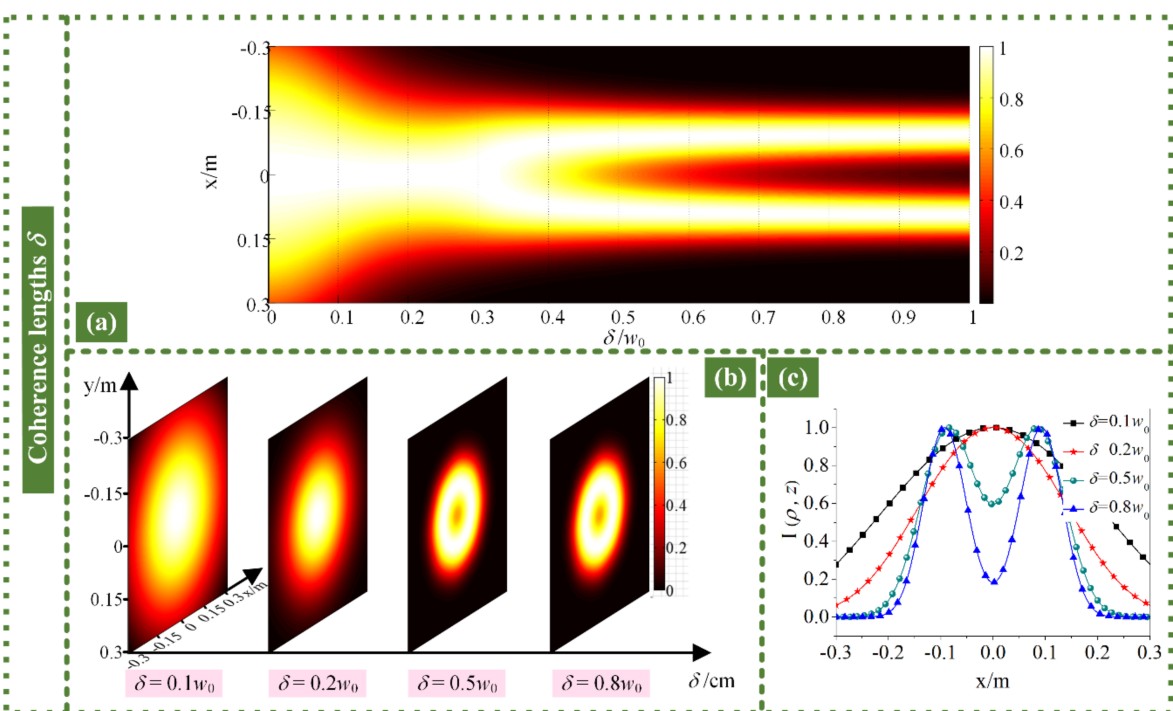

**Figure 6.** (**a**) Evolution of intensity distributions of RPLPCV beam array through non-Kolmogorov turbulence with coherence lengths $\delta$; intensity (**b**) distributions and (**c**) profiles of the RPLPCV beam array propagated through non-Kolmogorov turbulence for different coherence lengths.

Figure 5 explores the influence of different wavelengths and waist radii and coherence lengths on the intensity of the RPLPCV beam array propagated through non-Kolmogorov turbulence in the two ways of distributions and profiles. It can be seen from Figure 5a1,b1 that the larger the wavelength and waist radius are, the dark area in the center of the intensity distribution becomes smaller and gradually disappears. Figure 5a2,b2 more intuitively shows that as the wavelength and waist radius increase, the center-point intensity increases. Therefore, when the wavelength and beam waist radius are larger, the propagation distance required for the intensity distribution of the RPLPCV beam array to be converted into a Gaussian distribution in the non-Kolmogorov turbulence is shorter. In other words, the longer the wavelength is and the larger the waist radius is, the greater the influence of atmospheric turbulence on the intensity distribution of the RPLPCV beam array. In addition, when the waist radius of a single partially coherent vortex beam increases, under the same conditions, the overlapping part of the intensity of adjacent beams increases. This is also one of the reasons why the intensity distribution rapidly approaches the Gaussian distribution.

The coherence length is also an important source parameter of the RPLPCV beam array. Figure 6a shows the evolution pattern of the intensity distribution of the RPLPCV beam array propagated through non-Kolmogorov turbulence along with the coherence length. Figure 6b,c show the intensity distribution and curve, respectively, when the special coherence length $\delta = 0.1w_0$, $0.2w_0$, $0.5w_0$ and $0.8w_0$ in Figure 6a. We can find that the smaller the coherence length is, the closer the intensity distribution of the RPLPCV beam array is to the Gaussian distribution after the non-Kolmogorov turbulence. For a smaller coherence length, the intensity distribution is obviously expanded after being converted into a Gaussian distribution. That is to say, under the same conditions, the smaller the coherence length is, the more the RPLPCV beam array will be affected by non-Kolmogorov turbulence. When the coherence length increases, the change in the intensity distribution of the RPLPCV beam array propagated after non-Kolmogorov turbulence is smaller. In other words, the larger coherence length has less influence on the intensity distribution. As

can be seen from Figure 6c, as the coherence length increases, the center-point intensity gradually decreases.

### 3.3. Analysis of the Influence of Atmospheric Turbulence Parameters

The parameters that characterize atmospheric turbulence in the non-Kolmogorov spectrum model include generalized exponent factor $\alpha$, atmospheric refractive index structure constant $C_0$, inner scale $l_0$ and outer scale $L_0$. The effects of turbulence parameters on the intensity distribution of the RPLPCV beam array are shown in Figures 7–9.

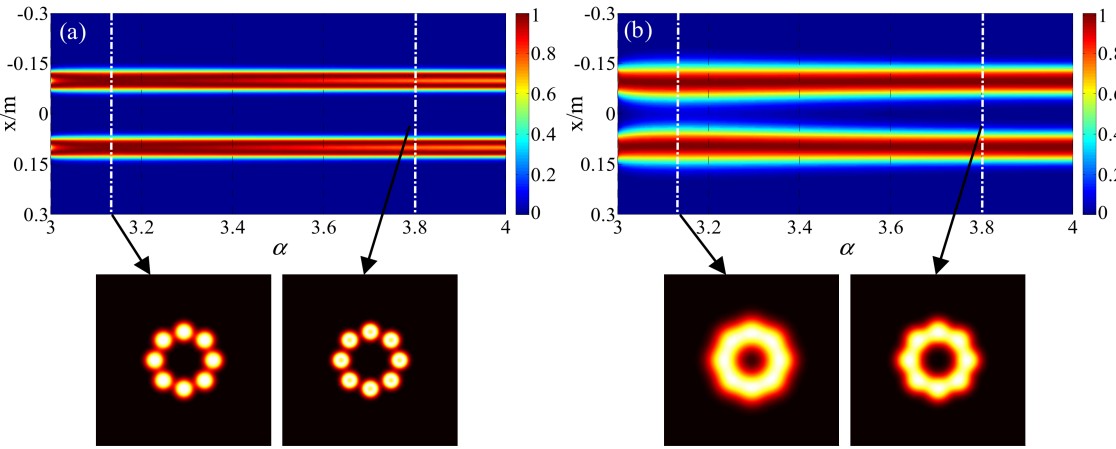

**Figure 7.** Evolution of intensity distribution of RPLPCV beam array through non-Kolmogorov turbulence versus generalized exponent factor $\alpha$ (**a**) $z = 1$ km; (**b**) $z = 2$ km.

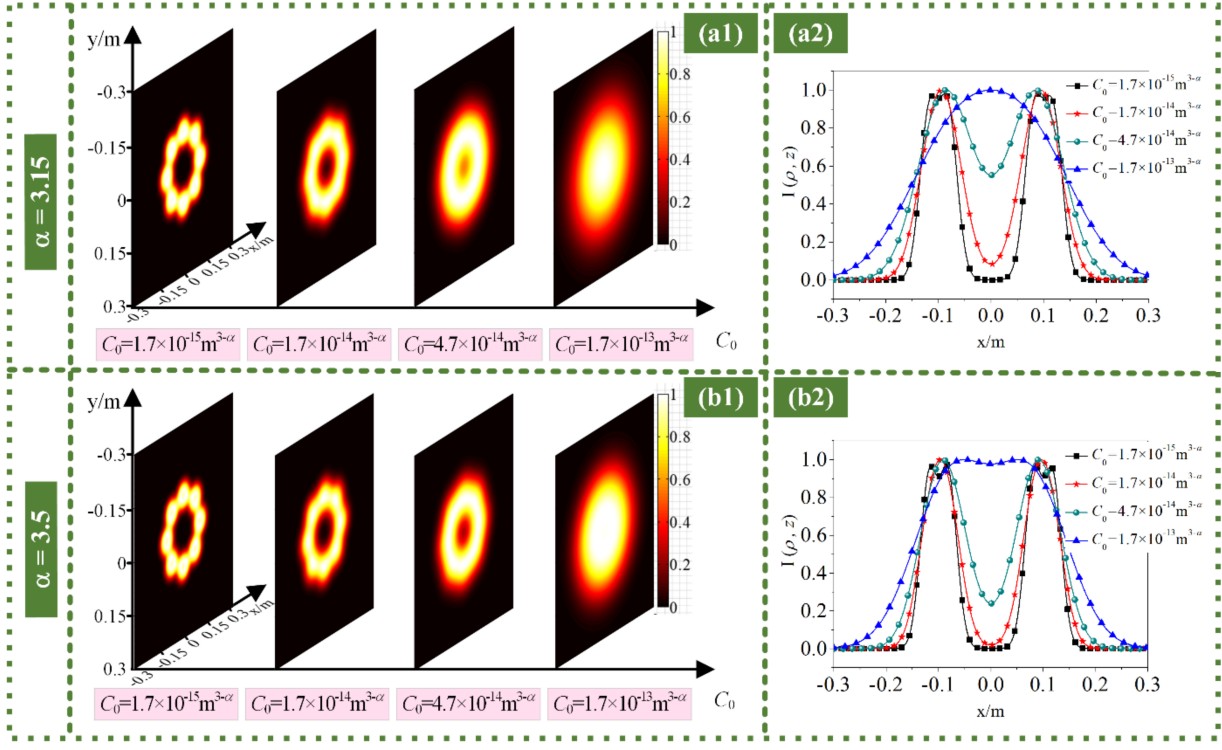

**Figure 8.** Intensity (**a1,b1**) distributions and (**a2,b2**) profiles of RPLPCV beam array through non-Kolmogorov turbulence for different $C_0$ values with generalized exponent factors (**a**) $\alpha = 3.15$ and (**b**) $\alpha = 3.5$.

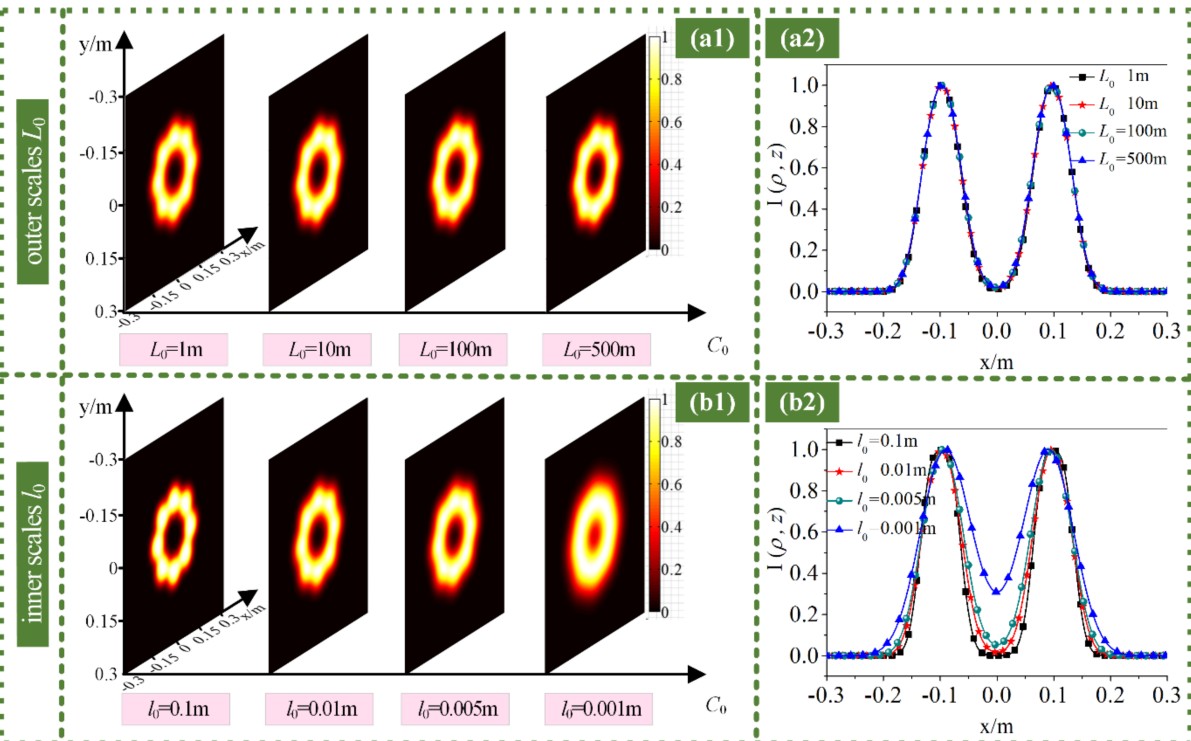

**Figure 9.** Intensity (**a1**,**b1**) distributions and (**a2**,**b2**) profiles of RPLPCV beam array through non-Kolmogorov turbulence for different (**a**) outer scales $L_0$ and (**b**) inner scales $l_0$.

Figure 7 demonstrates the evolution pattern of the intensity distribution of the RPLPCV beam array propagated through non-Kolmogorov turbulence along with the generalized exponential factor $\alpha$. The generalized exponent factors corresponding to the two sets of intensity distribution diagrams are $\alpha = 3.15$ and $\alpha = 3.8$. It can be seen from Figure 7 that when the generalized exponent factor $\alpha \approx 3.15$, the RPLPCV beam array is most affected by the non-Kolmogorov turbulence. This is because the generalized exponent factor $\alpha \approx 3.15$ represents the maximum intensity of atmospheric turbulence. In addition, when the propagation distance is larger, the influence of the generalized exponent factor on the intensity distribution of the RPLPCV beam array propagated through non-Kolmogorov turbulence is more obvious.

Figure 8 shows the intensity distributions and profiles of the RPLPCV beam array propagated by non-Kolmogorov turbulence with different atmospheric refractive index structure constants $C_0$ with the generalized exponent factors $\alpha = 3.15$ and $\alpha = 3.5$. As can be seen from Figure 8, when the atmospheric refractive index structure constant is larger, the intensity distribution of the RPLPCV beam array propagated through non-Kolmogorov turbulence is expanded more severely. This also verifies that the larger the atmospheric refractive index structure constant is, the smaller the propagation distance required to convert the intensity distribution of the RPLPCV beam array into a Gaussian distribution. The reason for this phenomenon is that the larger the atmospheric refractive index structure constant, the stronger the turbulence intensity, and the greater the influence of atmospheric turbulence on propagation properties of beam, the more serious the expansion. In addition, for the same atmospheric refractive index structure constant, when $\alpha = 3.15$, the RPLPCV beam array is most affected by non-Kolmogorov turbulence. The reason for this phenomenon is illustrated in Figure 7.

Figure 9 explores the influence of different outer scales $L_0$ and inner scales $l_0$ on the intensity distribution of the RPLPCV beam array propagated through non-Kolmogorov turbulence. It can be found from Figure 9 that the outer scale has almost no effect on the intensity distribution of the RPLPCV beam array propagated through non-Kolmogorov

turbulence, as displayed in Figure 9a. However, as the inner scale becomes smaller, the intensity distribution of the RPLPCV beam array expands, as shown in Figure 9b. The main reason for this phenomenon is that the smaller the inner scale is, the more severe the diffraction of the beam in atmospheric turbulence is, and the random distribution of the beam in time and space results in a relatively scattered intensity distribution. Therefore, when studying the influence of the inner and outer scales of non-Kolmogorov turbulence on the intensity properties of the RPLPCV beam array, the outer scale $L_0$ can be ignored, and the larger the inner scale $l_0$ is, the smaller the influence of the non-Kolmogorov turbulence is. This is consistent with the influence of the inner and outer scales of non-Kolmogorov turbulence on a single vortex beam.

## 4. Conclusions

The beam array is an effective way to increase the power of the light source, and can increase the propagation distance of the FSO communication system. In this paper, taking the RPLPCV beam array with topological charge $l = 1$ as the research object, the intensity expression of the RPLPCV beam array propagated through non-Kolmogorov turbulence is derived, and the effects of the source parameters and non-Kolmogorov turbulence parameters on its propagation properties are discussed. We found that the intensity distribution of the RPLPCV beam array propagated in the non-Kolmogorov turbulence is first converted from the array beam into a ring beam, then gradually converted into a Gaussian-like beam, and finally converted into a standard Gaussian beam form. The larger the radial radius, radial number and waist radius, the smaller the coherence length, and the longer the wavelength is, the shorter the propagation distance required for the intensity distribution of the RPLPCV beam array to be converted into a Gaussian distribution is in the non-Kolmogorov turbulence. When the generalized exponent factor of non-Kolmogorov turbulence $\alpha = 3.15$, the non-Kolmogorov turbulence has the greatest influence on the RPLPCV beam array, and the influence of the outer scale is negligible, and the larger the inner scale is, the smaller the influence.

In summary, the research results of this thesis provide a theoretical reference value for the further use of partially coherent vortex beams to transmit information, and also provide a technical support for the reasonable selection of light source parameters. In addition, what is worthy of further study is the influence of atmospheric turbulence on forth-order propagation properties of the RPLPCV beam array, such as intensity scintillation, beam drift and $M^2$ factor.

**Author Contributions:** Methodology, J.W.; data curation, M.W. and Z.T.; formal analysis, J.W. and M.W.; writing—original draft preparation, J.W., M.W., Z.T., S.L., Y.W. and C.W.; writing—review and editing, J.W. and S.L.; project administration, J.W. and Y.W. All authors have read and agreed to the published version of the manuscript.

**Funding:** This research was funded by the National Natural Science Foundation of China (NSFC) (grant numbers 62101313, 62001363, 61771385); the Natural Science Foundation of Shaanxi Province, China (grant number 2019JQ-901); the Natural Science Foundation of Shaanxi Provincial Department of Education (grant number 20JK0690); Xi'an Science and Technology Planning Project (grant number 2020KJRC0038); and Beilin District 2020 Science and Technology Plan Project (grant number GX2009).

**Institutional Review Board Statement:** Not applicable.

**Informed Consent Statement:** Not applicable.

**Data Availability Statement:** Not applicable.

**Acknowledgments:** The authors express their appreciation to the anonymous reviewers for their valuable suggestions.

**Conflicts of Interest:** The authors declare no conflict of interest.

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
