# Peer review of "Influence of Source Parameters and Non-Kolmogorov Turbulence on Evolution Properties of Radial Phased-Locked Partially Coherent Vortex Beam Array"

_photonics, doi:10.3390/photonics8110512_

Round 1

Reviewer 1 Report

The paper by Wang et al. presented a theoretical analysis of radial phased-locked partially coherent vortex (RPLPCV) beam array on propagation in non-Kolmogorov turbulence, and discussed the influence of source parameters and atmospheric turbulence parameters on its intensity distribution. The main finding is that the intensity distribution of the array is gradually turn into a Gaussian distribution, which is not surprising. The result of the work seems correct to me, but the significance of the results does not meet the requirements of photonics. Moreover, In the abstract and a number of times in the paper, the authors suggest that the work will have "applications in wireless optical communication". This is a vague statement that does not justify publication in photonics unless it is backed up with a demonstration, or at the very least specific application ideas.

In addition to the basic concern about the novelty/impact of the paper, I have a few comments:

  1. After Eq.1, the authors give the theoretical model of radial phased-locked beam array in Eq.2. I wonder if the model is firstly introduced in this paper, and if so, the authors should derive the beam realization conditions of the new model. If the model has been introduced before, related references are needed.
  2. The authors discuss the situations for topological charge l=1, besides, whether the topological charge can be set other values?

Reviewer 2 Report

The paper called Evolution properties of radial phased-locked partially coherent  vortex beam array in non-Kolmogorov turbulence JiaoWang, Mingjun Wang, Sichen Lei, Zhenkun TAN , Chenbai Wang and Yuanfei Wang and the state of the literature included in it, the presented approach is quite original. The paper is very good and interesting. The main idea is clearly explained and I’m really impressed by the variety of research methods.

There are some major aspects I would like to highlight.

  • I suggest that the title of the publication be rewritten.
  • The paper should clearly emphasize what is the purpose and scope of the work.
  • The entire text should be checked for grammar.
  • What is the future direction of the development of the radial phased-locked…?
  • The publication is not finished, the conclusion is rather a continuation of the discussion.
  • It is useful to include information in the summary as to how the publication contributes to science.

The presented conclusions may be of fundamental importance, therefore they should be presented in a better light and the author’s should emphasize the original research contribution. I believe, that suggested amendments will significantly increase the relevance of the publication and will improve it. After applying all required changes, the paper is suitable for publication.

Reviewer 3 Report

This manuscript presents a computational study reporting the evolution properties of radial phased-locked partially coherent vortex beam array in non-Kolmogorov turbulence. The authors reported analytical expressions, and the evaluated numerical results for the cross-spectral density and the average intensity of a radial phased-locked partially coherent vortex beam array propagated through non-Kolmogorov turbulence. This study has potential applications in facilitating free-space optical communications by providing a theoretical reference for selecting light sources and the suppression of turbulence effects. The work presented in this manuscript is interesting and a significant contribution to the field and the community. Furthermore, this manuscript is well written and credibly conveys the findings. Therefore, I have no hesitation in recommending this manuscript to be accepted for publication.

Author Response

Thank you for the reviewer’s comments and affirmation concerning our article. Those comments are all valuable and very helpful for revising and improving our article. We tried our best to improve the manuscript and made some changes in the manuscript, based on the comments of all reviewers. These changes will not influence the content and framework of the paper. We appreciate for the reviewers’ warm work earnestly, and hope that the correction will meet with approval. Once again, thank you very much for your comments and suggestions.

Round 2

Reviewer 1 Report

accept

Reviewer 2 Report

Thank you very much for including the amendments.

Now the article is ready for publication.